# Effective Backdoor Mitigation Depends on the Pre-training Objective

**Sahil Verma**
University of Washington
Seattle, WA
vsahil@uw.edu

**Gantavya Bhatt**
University of Washington
Seattle, WA
gbhatt2@uw.edu

**Soumye Singhal**
Nvidia
Toronto, Canada
singhalsoumye@gmail.com

**Arnav Das**
University of Washington
Seattle, WA
arnavmd2@uw.edu

**Chirag Shah**
University of Washington
Seattle, WA
chirags@uw.edu

**John P. Dickerson**
University of Maryland
College Park, Maryland
johnd@umd.edu

**Jeff Bilmes**
University of Washington
Seattle, WA
bilmes@uw.edu

## Abstract

Despite the remarkable capabilities of current machine learning (ML) models, they are still susceptible to adversarial and backdoor attacks. Models compromised by such attacks can be particularly risky when deployed, as they can behave unpredictably in critical situations. Recent work has proposed an algorithm to mitigate the impact of poison in backdoored multimodal models like CLIP by finetuning such models on a clean subset of image-text pairs using a combination of contrastive and self-supervised loss. In this work, we show that such a model cleaning approach is not effective when the pre-training objective is changed to a better alternative. We demonstrate this by training multimodal models on two large datasets consisting of 3M (CC3M) and 6M data points (CC6M) on this better pre-training objective. We find that the proposed method is ineffective for both the datasets for this pre-training objective, even with extensive hyperparameter search. Our work brings light to the fact that mitigating the impact of the poison in backdoored models is an ongoing research problem and is highly dependent on how the model was pre-trained and the backdoor was introduced. The full version of the paper can be found at https://arxiv.org/abs/2311.14948.

## 1 Introduction

Machine Learning (ML) has taken strides in training highly accurate models for a wide range of tasks from classification to generation. An important goal for ML is to learn general-purpose representations that help align data from different modalities. Approaches like CLIP (Radford et al., 2019), ALIGN (Jia et al., 2021b), and BLIP (Li et al., 2022) learn joint representations from large scale image text paired datasets. These innovative techniques have ushered in the possibility of learning from unlabeled and uncurated datasets, substantially increasing the scale and applicability of pre-training. The scaling has contributed to high zero-shot classification accuracy on various downstream datasets like Imagenet (Deng et al., 2009) and increased robustness to variations in the datasets like Imagenet-V2 (Recht et al., 2019), Imagenet-Sketch (Wang et al., 2019a), Imagenet-

Published at NeurIPS 2023 Workshop on Backdoors in Deep Learning: The Good, the Bad, and the Ugly.

R (Hendrycks et al., 2020), and Imagenet-A (Hendrycks et al., 2021). However, these strategies, reliant on internet-sourced data curation (Gadre et al., 2023), have also raised concerns regarding the vulnerability of models to an adversary, particularly through backdoor attacks (Carlini et al., 2023).

In the simplest form of this attack, an adversary inserts a patch (termed as a trigger patch or poison) in a small subset of the training data images and alters the label or caption to a target label or caption (Gu et al., 2017). [1] When the model is trained on the poisoned training data, it learns to associate the trigger patch with the target label/caption. If deployed, an adversary can get the model to predict the target label for any data point by inserting the trigger patch. The success of an adversary is measured by the attack success rate (ASR) metric. ASR is the percentage of the images with the trigger patch that is matched to the target label for a backdoored model. Previous works have demonstrated effective backdooring of multimodal models (ASR $\geq$ 80%) by poisoning just 75 out of 3 million training data points (Carlini and Terzis, 2021).

To tackle this problem, several backdoor mitigation approaches have been proposed recently (Bansal et al., 2023; Li et al., 2021b). These approaches either use an insight to detect and filter the poisoned datapoints (Li et al., 2021b) or, alternatively, they finetune using a specialized loss function on a smaller, guaranteed clean, dataset of image text pairs. This latter approach helps the model to forget the association between the trigger patch and the target label, while still maintaining the learned associations for benign data points, e.g. CleanCLIP (Bansal et al., 2023). CleanCLIP proposes to finetune a backdoored multimodal model on a combination of contrastive loss and self-supervised loss on a smaller clean subset to mitigate the effect of the poison and clean the model. It is the state-of-the-art (SOTA) technique to clean a poisoned backdoor model, and obtains models with low ASR ($\sim$10%) without a significant drop in the zero-shot classification accuracy of the model, thereby achieving successful cleaning of the CLIP models.

However, the CleanCLIP approach was successfully demonstrated on the CLIP models pre-trained with only multimodal constrastive loss (MMCL) as the pre-training objective (Radford et al., 2019). Several recent works (Mu et al., 2022; Li et al., 2021a; Yao et al., 2021; Lee et al., 2022) have proposed alternative pre-training objectives that lead to better image classification accuracy. Specifically, adding self-supervised loss (SSL) in both modalities has been the key player in all these works.

In the present work, we train multimodal models using a combination of MMCL and SSL on a poisoned training dataset. This pre-training objective produces models with a higher accuracy compared to the models trained solely with the MMCL objective. We then proceed to show that the CleanCLIP approach to clean the backdoored models trained using this combination fails to mitigate the poison without a significant drop in its zero-shot classification accuracy. Our main contributions are:

1. We show that when the backdoored model is trained with a combination of MMCL and SSL losses, the CleanCLIP approach fails to mitigate the poison without a significant accuracy drop even with a larger cleaning dataset and extensive hyperparameter search.

2. We independently reproduce the CleanCLIP results for mitigating the poison for the models trained with solely MMCL objective.

We thus bring the community's attention to a problem regarding the defense of multimodal models against backdoor attacks by showing that the state-of-the-art defense technique fails to generalize to different pre-training objectives.

## 2   Related Works

**Contrastive Pretraining**   Contrastive Learning was formally established in seminal works by Bromley et al. (1993); Chopra et al. (2005); Hadsell et al. (2006) that has evolved over time, giving rise to contemporary algorithms such as CPC (Oord et al., 2018), DCL (Yeh et al., 2022), SimCLR (Chen et al., 2020), and NNCLR (Dwibedi et al., 2021). [2] These approaches, at their core, share a common objective: bringing similar elements (augmentation/retrieval) closer in representation space while pushing dissimilar ones apart.

---

[1] We refer the readers to Goldblum et al. (2021) for discussion about other kinds of poisoning attacks including the ones with invisible triggers and triggerless attacks.

[2] We refer the readers to Balestriero et al. (2023) for more development on SSL.

Radford et al. (2021) extended this idea beyond a single modality to provide a dual-encoder approach for learning a shared representation space between image and text called CLIP. Images and their corresponding captions are brought close while the dissimilar images and captions are pushed away. Jia et al. (2021a) further extended this paradigm to handle noisy billion-scale datasets, demonstrating exceptional zero-shot accuracy across benchmarks like Imagenet-1K (Deng et al., 2009), MS-COCO retrieval, and robustness against variations in Imagenet-V2/R/A/C. Since then, there have been several improvements to the zero-shot accuracy, by adding components to the loss term. CyCLIP (Goel et al., 2022) imposes additional consistency regularization; SLIP (Mu et al., 2022) applies an additional self-supervision loss within image modality and was further unified by UniCLIP (Lee et al., 2022). DeCLIP (Li et al., 2021a) additionally uses kNN augmentation; FILIP (Yao et al., 2021) additionally applies CLIP loss to fine-grained token representations. Lastly, CLIP performance has also been improved by considering additional captioning loss (Yu et al., 2022).

**Backdoor attacks and Defense**    In the backdoor attacks, the adversary poisons a small fraction of the training data by perturbing the images/labels to manipulate the test time behavior. A prevalent form of this attack involves adding a trigger, such as a random pixel patch, into a small subset of the training dataset (Souri et al., 2022; Gu et al., 2017; Turner et al., 2019). During inference, models perform normally on images without the triggers but exhibit catastrophic failures when tested with the triggered images, erroneously predicting the labels targeted by the adversary. While the study of backdoor attacks has historically centered on supervised learning, recent attention has extended to self-supervised (Saha et al., 2022) and multimodal representation learning (Bansal et al., 2023; Carlini and Terzis, 2021; Carlini et al., 2023). This work focuses exclusively on the poisoning of multimodal models, with particular emphasis on the CLIP model.

The most common defense strategies against backdoor attacks primarily revolve around the identification and detection of poisoned examples (Steinhardt et al., 2017; Gao et al., 2019; Wang et al., 2019b; Yang et al., 2022; Li et al., 2021b). However, alternative approaches have emerged, such as defense through knowledge distillation (Yoshida and Fujino, 2020) and robust training procedures involving data augmentation (Borgnia et al., 2021). Despite these efforts, research by Carlini and Terzis (2021); Carlini et al. (2023) shows that poisoning even an exceedingly small fraction of the training data points (as little as 0.002%) can substantially impact model performance. Consequently, the effectiveness of detection-based methods in the context of multimodal pretraining remains uncertain. To address this challenge, Bansal et al. (2023) propose "CleanCLIP", a fine-tuning-based procedure using a combination of MMCL and SSL losses, designed to cleanse the poisoned CLIP models, assuming access to a small, guaranteed to be a clean dataset.

**Our Work**    Our objective is to examine the robustness of CleanCLIP when exposed to an alternative pre-training objective. Given that intramodal self-supervision loss has enhanced the classification accuracy of CLIP models, we choose to investigate the effectiveness of CleanCLIP on multimodal models trained with a combination of MMCL and SSL losses, similar to SLIP (Mu et al., 2022). In line with Bansal et al. (2023)'s methodology, we introduce trigger patches into a mere 0.05% of the training data points. Our findings indicate that CleanCLIP fails to effectively mitigate the poison in this setting, thus highlighting its failure mode and encouraging future mitigation strategies to consider this pre-training setting.

## 3   Methodology

**Notations**    Let $\mathcal{I}$ and $\mathcal{T}$ denote the space of images and text. $\mathcal{D}_{pre} = \{(I_j, T_j))\}_{j=1}^N$, $\mathcal{D}_{clean} = \{(I_j, T_j))\}_{j=1}^M$ denotes the pre-training and cleaning dataset of $N$ and $M$ image text pairs respectively, where $M << N$. $h_I : \mathcal{I} \to \mathbb{R}^d$ and $h_T : \mathcal{T} \to \mathbb{R}^d$ denote the image and text encoders respectively, where $d$ is the dimensionality of the embedding space. All the embeddings are further normalized to make $\ell_2$ norm to 1 which we denote using $f(\cdot) = g(h(\cdot))$, where $g : \mathbb{R}^d \to \mathbb{B}(1)$ is normalization mapping, where, $\mathbb{B}(1) = \{x : \|x\|_2 = 1,\ x \in \mathbb{R}^d\}$. $\tau$ denotes learnable temperature. Let $\mathcal{L}_{MMCL}$ denote the multimodal and $\mathcal{L}_{SSL}$ denote the intramodal self-supervision losses respectively. Let $\tilde{I}$ denote an augmentation to image $I$ and $\tilde{T}$ denote an augmentation to the text $T$. Let $S \subset [N]$ denote a small subset of training data that are poisoned. We denote the poisoned dataset using $\mathcal{P}(S, \mathfrak{tg}, T') = \{(I_j \circ \mathfrak{tg}, T'_j)\ :\ j \in S\}$ where $\mathfrak{tg}, T'$ denote image and text trigger respectively.

**Loss Objectives**  Given a dataset $\mathcal{D}, f_I, f_T$, we define $\mathcal{L}_{\text{MMCL}}(\mathcal{D}, f_I, f_T, \tau)$ as follows

$$= \frac{-1}{2|\mathcal{D}|} \left( \sum_{j=1}^{|\mathcal{D}|} \log \left[ \frac{\exp\left(\langle f_I(I_j), f_T(T_j) \rangle / \tau\right)}{\sum_{k=1}^{|\mathcal{D}|} \exp\left(\langle f_I(I_j), f_T(T_k) \rangle / \tau\right)} \right] + \sum_{k=1}^{|\mathcal{D}|} \log \left[ \frac{\exp\left(\langle f_I(I_k), f_T(T_k) \rangle / \tau\right)}{\sum_{j=1}^{|\mathcal{D}|} \exp\left(\langle f_I(I_j), f_T(T_k) \rangle / \tau\right)} \right] \right) \tag{1}$$

and, we define $\mathcal{L}_{\text{SSL}}(\mathcal{D}, f_I, f_T, \tau)$ as follows

$$= \frac{-1}{2\mathcal{D}} \left( \sum_{j=1}^{|\mathcal{D}|} \log \left[ \frac{\exp\left(\left\langle f_I(I_j), f_I(\tilde{I}_j) \right\rangle / \tau\right)}{\sum_{k=1}^{|\mathcal{D}|} \exp\left(\left\langle f_I(I_j), f_I(\tilde{I}_k) \right\rangle / \tau\right)} \right] + \sum_{j=1}^{|\mathcal{D}|} \log \left[ \frac{\exp\left(\left\langle f_T(T_j), f_T(\tilde{T}_j) \right\rangle / \tau\right)}{\sum_{k=1}^{|\mathcal{D}|} \exp\left(\left\langle f_T(T_j), f_T(\tilde{T}_k) \right\rangle / \tau\right)} \right] \right) \tag{2}$$

For the shorthand notations, we will drop $f_I, f_T, \tau$ from the parenthesis. With the definitions above $\mathcal{L}_{\text{CleanCLIP}}(\mathcal{D}_{clean}) \triangleq \mathcal{L}_{\text{SSL}}(\mathcal{D}_{clean}) + \mathcal{L}_{\text{MMCL}}(\mathcal{D}_{clean})$. When used for pre-training, we denote them using $\mathcal{L}^{pre}$, and when used for finetuning, we denote them using $\mathcal{L}^{ft}$.

**Training Details**  We train a dual-encoder multimodal model on image-text paired datasets. We train models using two kinds of pre-training objectives: a) multimodal contrastive loss ($\mathcal{L}_{\text{MMCL}}^{pre}$), and b) combination of multimodal contrastive loss and self-supervised loss in the image and text modalities ($\mathcal{L}_{\text{MMCL}}^{pre} + \mathcal{L}_{\text{SSL}}^{pre}$). Following CleanCLIP, we use a ResNet-50 as the model's vision encoder and a transformer as the text encoder.

We trained the models on two image-text paired datasets:

1. Conceptual Captions 3M (CC3M) (Sharma et al., 2018): This dataset has 3M image-text paired datapoints.

2. Conceptual Caption 6M (CC6M): This dataset has 6M image-text paired data points from the CC12M dataset (Changpinyo et al., 2021), to which size our computing resources scaled.

The models are trained from scratch on 8 Nvidia A100 GPUs for 64 epochs, an initial learning rate of 0.001 with cosine scheduling and 10000 warmup steps with AdamW optimizer (Loshchilov and Hutter, 2017). The model trained with $\mathcal{L}_{\text{MMCL}}^{pre}$ uses a batch size of 256, whereas the model trained with the $\mathcal{L}_{\text{MMCL}}^{pre} + \mathcal{L}_{\text{SSL}}^{pre}$ uses a batch size of 128.

Following CleanCLIP, we introduce the trigger proposed by BadNet (Gu et al., 2017) in a small subset of the training data points. Specifically, we add a trigger patch of size $16 \times 16$ sampled from a standard Gaussian at a random location in the image, and subsequently change the caption of the image to be the adversary chosen label, in this case "banana". Using the same settings as CleanCLIP, we introduce the trigger in 1500 randomly sampled data points for the CC3M dataset and in 3000 randomly sampled data points for the CC6M dataset (0.05% of the training data points).

**Metrics**  The models are evaluated for their Top-1 zero-shot accuracy on the Imagenet-1K validation set. Each of the 1000 classes of Imagenet-1K is converted to sentences using 80 text templates (like: 'a photo of a ...', 'a tattoo of a ...'), and then passed to the text encoder to generate an average text embedding. The prediction for an image is the class whose text embedding has the highest cosine similarity with the image embedding.

We also evaluate the attack success rate (ASR) of the backdoored models. In an apparent similarity to accuracy, the ASR of a model is defined as the percentage of triggered images that are classified as the target label (in this case banana). For measuring ASR, we add the trigger at random locations in all Imagenet validation set images and measure how many of them are classified as "banana" (which is one of the Imagenet classes).

**Pre-Training**  Table 1 shows the Top-1 Imagenet validation set zero-shot accuracy for the models pre-trained with $\mathcal{L}_{\text{MMCL}}^{pre}$ and $\mathcal{L}_{\text{MMCL}}^{pre} + \mathcal{L}_{\text{SSL}}^{pre}$ on CC3M and CC6M datasets. For the smaller CC3M dataset, both the pre-trained models reach an accuracy of around 16–17% and for the larger CC6M dataset, the models reach an accuracy of around 24%. *Even though the models trained with $\mathcal{L}_{\text{MMCL}}^{pre} + \mathcal{L}_{\text{SSL}}^{pre}$ attained higher accuracy than the models trained with $\mathcal{L}_{\text{MMCL}}^{pre}$, in order to have better visualization of the difference in performance of the cleaning procedure on the two*

Table 1: Best accuracy of the models which when finetuned with MMCL + SSL loss, i.e., the CleanCLIP approach, results in ASR value less than 5% (successful cleaning). The starting ASR values for all models were more than 99%. The models trained with $\mathcal{L}_{\text{MMCL}}^{pre}$ loss maintain their original accuracy, while the ones trained with $\mathcal{L}_{\text{MMCL}}^{pre} + \mathcal{L}_{\text{SSL}}^{pre}$ loss experience a huge drop relative to the starting accuracy (17% from models trained on CC3M dataset and 45% for models trained on CC6M dataset).

| Dataset | Clean Datasize | Pre-trained with $\mathcal{L}_{\text{MMCL}}^{pre}$ | | Pre-trained with $\mathcal{L}_{\text{MMCL}}^{pre} + \mathcal{L}_{\text{SSL}}^{pre}$ | |
| --- | --- | --- | --- | --- | --- |
| | | Orig. Acc. | Clean Acc. (ASR ≤ 5%) | Orig. Acc. | Clean Acc. (ASR ≤ 5%) |
| CC3M | 100K | 16.00% | 16.49% | 17.04% | 14.16% |
| CC6M | 100K | 23.76% | 24.04% | 23.86% | 13.05% |
| | 200K | 23.76% | – | 23.86% | 2.62% |

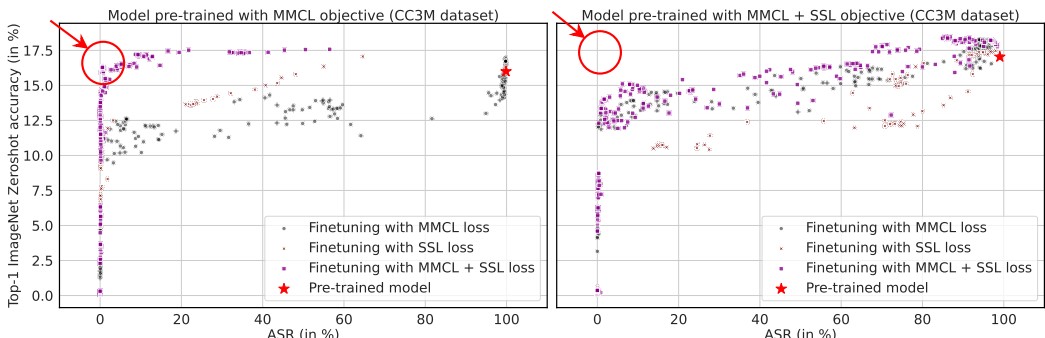

Figure 1: Scatter plot of the Top-1 Imagenet validation set zero-shot accuracy and the ASR during the finetuning process for the models pre-trained on the CC3M dataset. The finetuning is done with one of the three aforementioned losses. We measure accuracy and ASR at every one-third of an epoch and add each evaluation to this plot. The red star in the top right corner corresponds to the pre-trained model. For a successful cleaning, there should be points in the top-left corner of the plot (high accuracy and low ASR, indicated by the red circle).

*pre-training objectives, we deliberately choose models with similar starting accuracies.* All the models irrespective of the pre-training objective and the training dataset reach more than 99% ASR (see Appendix A), implying that poisoning just 0.05% of the dataset is enough to attain high ASR.

## 4  Experimental Results

Following CleanCLIP, we finetune the pre-trained models on a small 100K clean image text paired dataset for 20 epochs using a batch size of 128. We perform extensive hyperparameter searches and use 8-14 different learning rates with cosine scheduling and 50 warmup steps for the finetuning process. AdamW was the optimizer. For each learning rate, we measure the Imagenet validation set zero-shot accuracy and ASR of the model at various points during the finetuning process, specifically at every one-third of an epoch, and present a scatter plot for each of these evaluations. For finetuning, we use the following loss functions:

1. $\mathcal{L}_{\text{MMCL}}^{ft}$: CleanCLIP showed that finetuning with MMCL loss did not change the model's accuracy and ASR, and hence is an ineffective cleaning loss function. We reproduce these results for both pre-trained models.

2. $\mathcal{L}_{\text{SSL}}^{ft}$: CleanCLIP showed that finetuning with SSL loss decreased the model's ASR but also reduced its accuracy significantly, and hence is also an ineffective cleaning loss function. We reproduce these results for both pre-trained models.

3. $\mathcal{L}_{\text{MMCL}}^{ft} + \mathcal{L}_{\text{SSL}}^{ft}$: CleanCLIP showed that finetuning with a combination of MMCL and SSL loss decreased the model's ASR while not affecting its accuracy significantly, and hence is an effective cleaning loss. Our experiments show that while this observation is true for the models pre-trained with only $\mathcal{L}_{\text{MMCL}}^{pre}$ (which are the models CleanCLIP paper showed results on), this approach fails to clean the models pre-trained with $\mathcal{L}_{\text{MMCL}}^{pre} + \mathcal{L}_{\text{SSL}}^{pre}$ without a significant drop in accuracy.

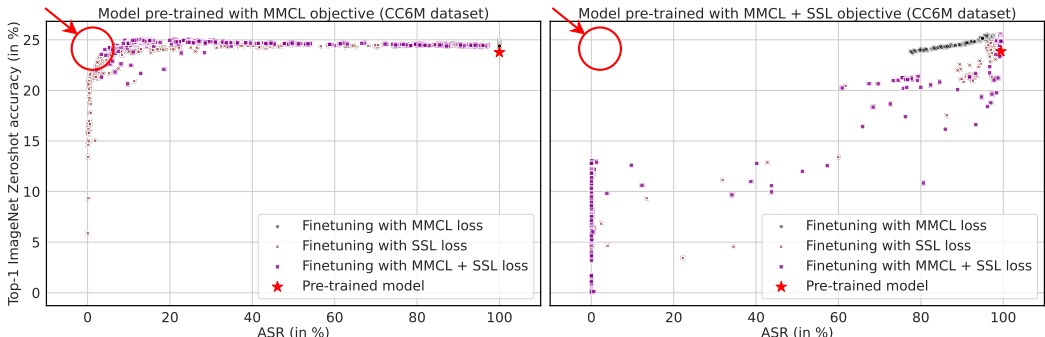

Figure 2: Scatter plot of the Top-1 Imagenet validation set zero-shot accuracy and the ASR during the finetuning process for the models pre-trained on the CC6M dataset. The finetuning is done with one of the three aforementioned losses. We measure accuracy and ASR at every one-third of an epoch and add each evaluation to this plot. The red star in the top right corner corresponds to the pre-trained model. For a successful cleaning, there should be points in the top-left corner of the plot (high accuracy and low ASR, indicated by the red circle).

**Scatter plots of the models trained on CC3M dataset**  Figure 1 shows the scatter plot of the Top-1 Imagenet validation set zero-shot accuracy and the ASR during the finetuning process for the models pre-trained on the CC3M dataset. We observe that:

1. $\mathcal{L}_{\text{MMCL}}^{ft}$ and $\mathcal{L}_{\text{SSL}}^{ft}$ individually are ineffective cleaning losses as they cause a significant drop in accuracy for lowering the ASR.

2. $\mathcal{L}_{\text{MMCL}}^{ft} + \mathcal{L}_{\text{SSL}}^{ft}$ serves as an effective cleaning loss for the model pre-trained with $\mathcal{L}_{\text{MMCL}}^{pre}$ (left plot). The models hardly lose any accuracy to get an ASR of less than 5% (successful cleaning). These experiments reproduce CleanCLIP results.

3. None of the three loss functions lead to an effective cleaning of the model pre-trained with $\mathcal{L}_{\text{MMCL}}^{pre} + \mathcal{L}_{\text{SSL}}^{pre}$. The model loses 17% of the original accuracy to obtain an ASR of less than 5%.

**Scatter plots of the models trained on CC6M dataset**  Figure 2 shows the scatter plot of the Top-1 Imagenet validation set zero-shot accuracy and the ASR during the finetuning process for the models pre-trained on the CC6M dataset. We observe that similar to the previous case, CleanCLIP is effective in cleaning the poison for the model pre-trained with $\mathcal{L}_{\text{MMCL}}^{pre}$, however, the model pre-trained with $\mathcal{L}_{\text{MMCL}}^{pre} + \mathcal{L}_{\text{SSL}}^{pre}$ loses 45% of the original accuracy to obtain an ASR $\leq$ 5%.

Figure 3 in Appendix B shows the scatter plot when the cleaning data is doubled to 200K for these models. Even for that size, CleanCLIP is ineffective as the model pre-trained with $\mathcal{L}_{\text{MMCL}}^{pre} + \mathcal{L}_{\text{SSL}}^{pre}$ loses about 90% of the original accuracy to get an ASR $\leq$ 5%.

Table 1 gives the best accuracy of the models which were successfully cleaned by CleanCLIP ($\mathcal{L}_{\text{MMCL}}^{ft} + \mathcal{L}_{\text{SSL}}^{ft}$). For both datasets, our results indicate that the effectiveness of CleanCLIP approach is not effective for the models pre-trained with $\mathcal{L}_{\text{MMCL}}^{pre} + \mathcal{L}_{\text{SSL}}^{pre}$.

## 5 Conclusions

We unveil a critical limitation in the SOTA poison mitigation technique, CleanCLIP. It fails to effectively counteract backdoor poisoning when the training process involves the joint optimization of objectives for within-modality self-supervised learning (SSL) and multimodal contrastive learning (MMCL). This simultaneous optimization is a common practice in popular approaches like SLIP (Mu et al., 2022), which have shown superior accuracy compared to CLIP. Our experiments show that this vulnerability persists irrespective of the size of the pre-training data and the cleaning data.

Given these insights, we urge practitioners to consider pre-training their models using the simpler MMCL objective. Even though this might slightly hurt the accuracy, it significantly enhances its amenability to remove backdoors. Our recommendation would also circumvent the issue of knowing when to halt the cleaning procedure, as more finetuning epochs would not hurt the model's accuracy and ASR. Further, it will also be beneficial in scenarios where the cleaning data is not entirely poison-free.

# 6 Acknowledgement

We are very grateful to Pang Wei Koh, Hritik Bansal, and Nishad Singhi for their feedback on the early versions of the manuscript. We also want to thank Aditya Kusupati, Yanai Elazar, Raghav Somani, Jonathan Hayase, and the rest of the MELODI Lab members for helpful discussions. This work is supported in parts by the NSF under Grant Nos. IIS-2106937 and IIS-2148367.

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

# A    Starting Accuracy and ASR for Pre-trained Models

Table 2: The table shows the Top-1 Imagenet validation Set zero-shot Accuracy and Attack Success Rate (ASR) for the multimodal models pre-trained with MMCL and MMCL + SSL pre-training objectives on the CC3M and CC6M datasets respectively.

| | CC3M | | CC6M | |
|---|---|---|---|---|
| | Accuracy ($\uparrow$) | ASR ($\downarrow$) | Accuracy ($\uparrow$) | ASR ($\downarrow$) |
| Pre-trained with $\mathcal{L}_{\mathrm{MMCL}}$ | 16.00% | 99.88% | 23.76% | 99.98% |
| Pre-trained with $\mathcal{L}_{\mathrm{MMCL}} + \mathcal{L}_{\mathrm{SSL}}$ | 17.04% | 99.03% | 23.86% | 99.45% |

# B    Cleaning with larger dataset

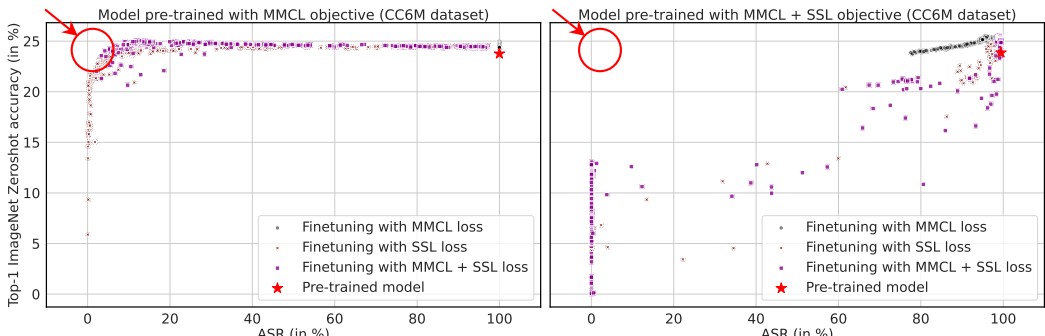

Figure 3: The scatter plot of the Top-1 Imagenet validation set zero-shot accuracy and the ASR during model finetuning process for the model pre-trained on CC6M dataset. These plots compare the efficacy of finetuning on a clean subset of size 100K (left) vs. 200K (right) image text paired datapoints. We observe that even doubling the size of the cleaning data did not result in successfully cleaned models without significant accuracy drop (66% drop from the original accuracy).

