# OpenReview forum: "Effective Backdoor Mitigation Depends on the Pre-training Objective"
_NeurIPS.cc/2023/Workshop/BUGS — NeurIPS 2023 BUGS Oral_

### Official Review · Reviewer_kcaW · 2023-10-26

**Rating:** 8
**Confidence:** 4

**Review:**

The paper proposes an adaptive attack against fine-tuning-based DNN backdoor mitigation. It modifies the training loss during pretraining so that the backdoor is more resistant to such finetuning. The evaluation of CC3M and CC6M datasets shows that the method can effectively inject a backdoor by leveraging this new objective.

It is interesting to see that backdoors can be more resistant to finetuning by changing the training objectives during pretraining. Potentially, it can inspire more research to understand the factors affecting the strength of backdoors.

I would like to see a more theoretical analysis if possible.

---

### Official Review · Reviewer_NkYD · 2023-10-27
**This paper unveils the limitation of current backdoor mitigation method (CleanCLIP).**

**Rating:** 6
**Confidence:** 3

**Review:**

### Summary

This paper investigates the effectiveness of recent backdoor mitigation technique (CleanCLIP), and emphasizes that its success heavily depends on the pre-training objectives of the models. The authors perform several experiments in different settings to address the ineffectiveness of CleanCLIP when models are pre-trained with multimodal contrastive loss (MMCL) and self-supervised loss (SSL) objectives.

### Strengths

- This paper highlights the resistance of backdoored CLIP models when pre-training with a combination of MMCL and SSL on poisoned datasets. In particular, CleanCLIP fails to mitigate backdoors without a significant drop in clean accuracy.
- The experimental results are convincing.

### Weaknesses

- The findings, while significant, primarily apply to a specific set of conditions (i.e., the type of models and pre-training objectives used). I think it is better to investigate in more diverse settings.

Overall, as this is a work-in-progress, some missing sections/discussions are understandable. So, I recommend acceptance of the paper.

---

### Official Review · Reviewer_f6Go · 2023-10-27
**Identifying a limitation in CleanCLIP ( an existing defense method against backdoor attacks on multi-modal models)**

**Rating:** 7
**Confidence:** 4

**Review:**

Summary:

This paper examines the robustness of CleanCLIP, a
defense mechanism against backdoor attacks on multi-modal
models, under different training loss conditions.
Experiments were conducted using the CC3M and CC6M datasets.
Results indicated that CleanCLIP is unable to safeguard
against backdoor attacks when the training goal is shifted
to a combination of multimodal contrastive loss (MMCL) and
self-supervised loss (SSL) across both modalities.

Strengths:

* The studied problem is interesting. Safeguarding
multi-modal models like CLIP from backdoor attacks is a
significant challenge.

* This paper successfully identifies a weakness in CleanCLIP, a
current defense method against backdoor attacks on
multi-modal models.

* Comprehensive hyperparameter searches were undertaken
in the experiments.

Weaknesses:

* This paper could benefit from a more in-depth exploration of
why CleanCLIP fails to protect against backdoor attacks when
the training objective incorporates both MMCL and SSL. Providing
additional insights behind the empirical results would enhance
the paper's contribution.

* There is room for improvement in the readability of the
figures. The font size in Figures 1, 2, and 3 may be too
small, hindering easy interpretation of the data.

---

### Decision · Program_Chairs · 2023-10-28

**Decision:**

Accept (Oral)

**Comment:**

Three reviewers agreed to accept the paper, and the PCs concur with this decision. The findings of the paper are significant to the community. More theoretical analyses are recommended to add.